# Defurnishing with X-Ray Vision:
# Joint Removal of Furniture from Panoramas and Mesh

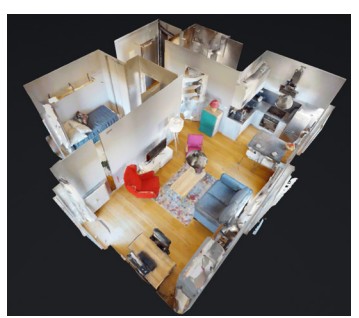 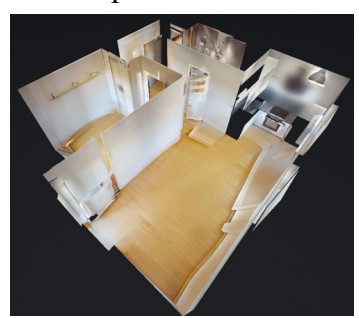 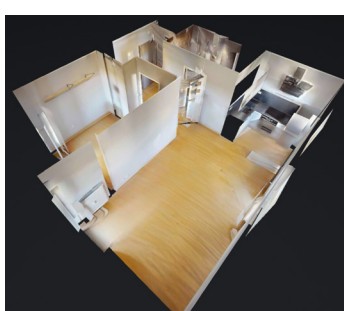

| Original mesh | Simplified defurnished mesh (SD, no control) | Simplified defurnished mesh (CN, control) |

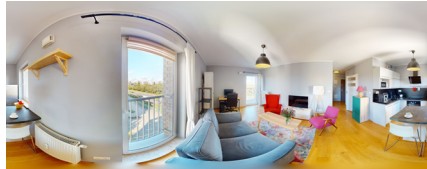 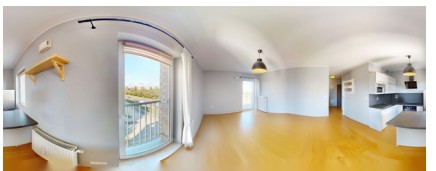 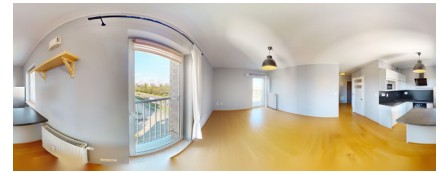

| Input 360 panorama image | Defurnished using SD (no mesh control) | Defurnished using CN with mesh control |

Figure 1. **Furniture removal based on simplified defurnished mesh (SDM).** We produce an SDM by removing furniture faces and closing holes in the input mesh. Then we render the SDM into depth and normal images, from which we extract Canny edges to use as structural guidance in ControlNet (CN) inpainting of the corresponding panorama images (right). Inpainting only with Stable Diffusion (SD) leads to warped lines in the output, such as those between walls and floor (middle).

## Abstract

We present a pipeline for generating defurnished replicas of indoor spaces represented as textured meshes and corresponding multi-view panoramic images. To achieve this, we first segment and remove furniture from the mesh representation, extend planes, and fill holes, obtaining a simplified defurnished mesh (SDM). This SDM acts as an "X-ray" of the scene's underlying structure, guiding the defurnishing process. We extract Canny edges from depth and normal images rendered from the SDM. We then use these as a guide to remove the furniture from panorama images via ControlNet inpainting. This control signal ensures the availability of global geometric information that may be hidden from a particular panoramic view by the furniture being removed. The inpainted panoramas are used to texture the mesh. We show that our approach produces higher quality assets than methods that rely on neural radiance fields, which tend to produce blurry low-resolution images, or RGB-D inpainting, which is highly susceptible to hallucinations.

## 1. Introduction

This paper presents a novel method for defurnishing 3D scenes represented as textured meshes and corresponding panoramic images. Defurnishing, the process of virtually removing furniture and clutter from a scene, has significant implications for real estate and digital twin applications. In real estate, defurnishing allows potential buyers to visualise a space without existing furniture, enabling virtual staging, and facilitating better property assessment. For digital twins, defurnishing provides a clean and uncluttered representation of a space, which is essential for tasks like facility management, space planning, and simulating renovations.

However, traditional defurnishing methods often struggle in scenes with heavy clutter, where the sheer volume

of objects can obscure the underlying structure of the space and lead to inaccurate furniture removal, inconsistent hole-filling, and artefacts in the associated 2D views. This is particularly problematic for applications like virtual staging, where realism and visual fidelity are paramount.

To address this challenge, we introduce a novel de-furnishing pipeline that leverages a *simplified defurnished mesh* (SDM) as a geometric prior. This simplified mesh, generated from the original scene, facilitates accurate and robust furniture removal, even in heavily cluttered environments. Furthermore, by combining the SDM with a ControlNet-based inpainting strategy, we ensure consistent and artefact-free results across both the 3D model and the 2D panoramic views. This combination of an SDM and ControlNet for defurnishing is a novel approach that allows us to overcome the limitations of existing methods.

Our approach offers several advantages. It excels in handling cluttered scenes, provides faster processing times compared to computationally intensive 3D-based inpainting methods, and adapts to diverse scenes due to its reliance on geometric priors rather than semantic segmentation. We demonstrate the effectiveness of our pipeline through extensive experiments on a diverse dataset of real-world 3D scenes, including those with significant clutter. Our results showcase superior performance in terms of visual quality, geometric accuracy, and consistency between the defurnished 3D model and 2D views. This work contributes to advancing 3D scene understanding and manipulation by providing a robust and efficient solution for defurnishing complex, real-world environments.

## 2. Related Work

The task of defurnishing requires furniture detection and removal. We use off-the-shelf semantic segmentation to identify furniture, so in this section we only review approaches that deal with object removal from images or scenes.

### 2.1. 2D Inpainting

Methods for single-image inpainting range from classical approaches [1, 3, 9, 17, 33, 48] to those leveraging neural networks, first pioneered by the use of generative adversarial networks [18, 34]. Subsequent improvements incorporate the attention mechanism [64, 67], adaptive convolutions [24, 65], fast Fourier convolutions [45], and image features such as edge maps [32] and semantic segmentation [44].

Latent diffusion models, such as Stable Diffusion (SD) [40], have recently risen to the forefront of image generation as they are readily scalable to model complex distributions of training data and can sample diverse inpaints at high fidelity [15, 25]. SD is a text-to-image model trained on a large image dataset [41] that can also be conditioned by multi-modal inputs, including line contours, depth maps, and other images for image-to-image translation [68]. SD has also been shown to be effective at removing objects in single images simply by fine-tuning on carefully curated datasets, without additional conditioning inputs [43, 61].

### 2.2. Multi-view Inpainting

Since the methods described above inpaint single images, when run on a set of images sharing visual overlap, such as for object removal, there is no guarantee that the inpaints will be consistent between images. One way to encourage *structural* consistency is to use ControlNet (CN) [68] to add conditioning to SD that is geometrically consistent across images, such as depth or normal vectors. However, this does not address *textural* consistency and is still susceptible to the same deviations and hallucinations as SD [50, 63], since controls are not guaranteed to be followed exactly.

To ensure multi-view consistency, the inpaints on single 2D images need to be propagated to other images through a 3D representation. Early methods use exemplar-based inpainting by evaluating reprojections from other views [22, 31, 36, 49], but perform poorly on larger masks and unobserved regions. Wei *et al*. [60] uses LaMa [45] to overcome these shortcomings and guarantees multi-view consistency via a novel iterative refinement process, while Ji *et al*. [20] use LaMa with panoramas. Other approaches exist without explicitly relying on a 3D representation, *e.g*. using attention mechanism [5].

Radiance fields, such as NeRF [27], can also be used for multi-view inpainting. Inpainting can be performed in 2D, and then used as input for optimising a NeRF in a multi-view consistent way [28, 29, 53, 59], but large inpainting regions lead to conflicting images and poor reconstructions. Following advancements in 3D generation by leveraging 2D diffusion priors, namely score distillation sampling [16, 23, 37, 51, 57], inpainting can also be performed jointly across all images [38, 58].

### 2.3. 3D Scene Inpainting

3D inpainting generally refers to completing missing parts of a 3D representation. Classical surface reconstruction approaches can only reliably fill small holes [21, 35], and there are learned approaches for point clouds [30, 42, 54] and signed-distance functions [11, 12]. Since we are primarily interested in applications where the 3D representation was reconstructed from 2D source data (*e.g*. posed images, depths, a video stream), this problem is intricately related to multi-view inpainting in 2D [13, 19]; performing inpainting on 2D images (and on other data needed for reconstruction, such as depth) in a multi-view consistent way can help in reconstructing the inpainted 3D scene. In fact, this is what all of the radiance field methods above do, since the multi-view consistent 2D inpaints are rendered from the radiance field representation of the scene.

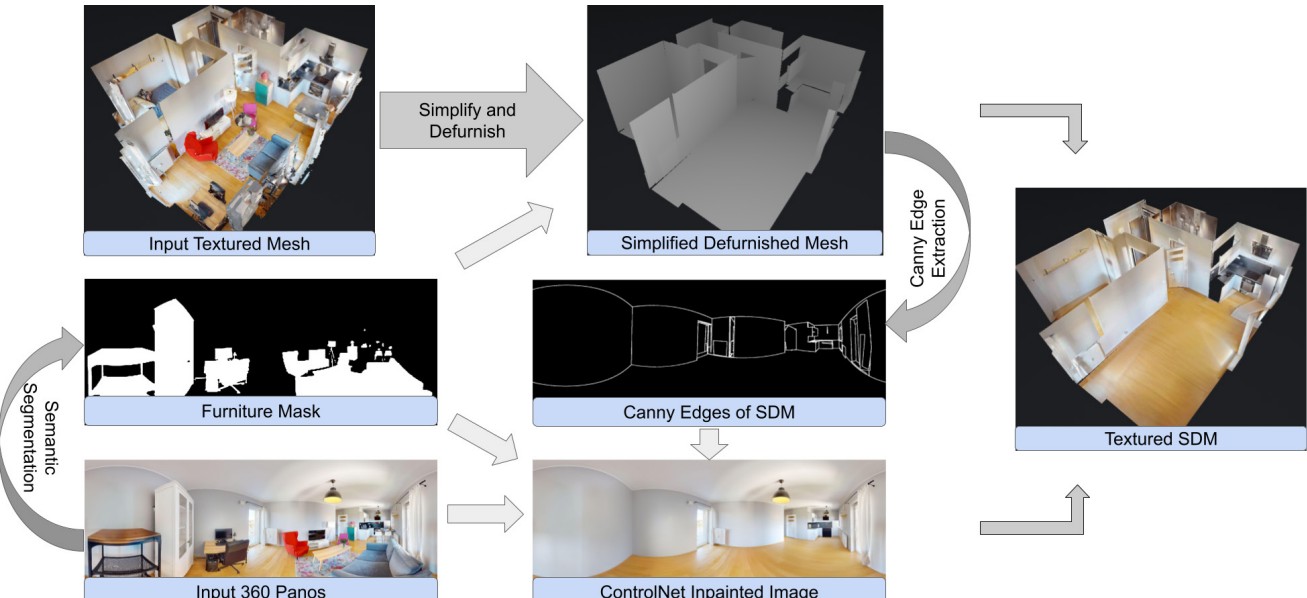

Figure 2. **Defurnishing pipeline overview.** Panoramic image segmentation guides simplification and defurnishing of an input textured mesh. Canny edges from the simplified mesh guide CN-based image defurnishing for final textured mesh reconstruction.

## 3. Method

This section details the defurnishing pipeline designed for 360° panorama images and a corresponding 3D textured mesh, reconstructed from these input images. The pipeline removes furniture from both the panoramas and the mesh, generating a complete defurnished scene. At a high level, our pipeline consists of 2D furniture segmentation, mesh simplification and defurnishing, CN inpainting, and texturing with the inpainted images, followed by super-resolution and blending of the final result. Figure 2 gives an overview.

### 3.1. Furniture Segmentation

For each 360° panorama image, a semantic segmentation model [7], trained on an ontology of common furniture, built-ins, and structural elements, is employed to classify the semantic category of each pixel. We use off-the-shelf training settings and a dataset of 20,000 equirectangular images, similar to ADE20K [62]. Based on the semantic segmentation results, specific categories corresponding to furniture items are selected. These categories are predefined and include common furniture types such as chairs, tables, sofas, and other free-standing furniture. This excludes structural elements, such as walls, floors, and ceilings, as well as built-ins and other objects not easily removable without tools. We also consider decorations and living beings (*i.e.* humans and animals) as furniture.[1] Given this furniture/non-furniture mapping, we generate a binary image, where pixels containing furniture are $true$. This mask

---

[1]All living beings are defurnished ethically.

is used as one of the inputs to the inpainter, as well as the mesh defurnishing pipeline. Please note that the masks do not cover shadows or reflections cast by any of the furniture.

### 3.2. Simplified Defurnished Mesh Generation

The objective for our SDM is to contain no furniture, while being structurally precise. Existing approaches [39, 52, 70] either over-simplify geometry or modify the placement of structures like walls, so we develop our own method. To generate the SDM, we first simplify the original textured mesh by approximating the scene's geometry with planar surfaces [2, 66], which facilitates efficient furniture removal, and hole-filling during the defurnishing process. An example of the output of this process is shown in Figure 2.

**Furniture Mask Projection**  The semantic segmentation masks, identifying furniture regions in the panoramas, are projected onto the input furnished mesh. This projection is achieved by leveraging the multi-view camera poses associated with the panorama images. This process effectively transfers and aggregates the 2D multi-view furniture masks onto the 3D mesh representation. The contributions of each of the multi-view furniture segmentation mask pixels are weighted by their distance from the observed faces.

**Mesh Defurnishing and Hole Filling**  Based on the projected labels of each mesh face, the faces representing furniture are removed from the simplified mesh. The resulting holes in the mesh are then filled by first projecting the removed faces to the nearest floor/wall plane, and then fill-

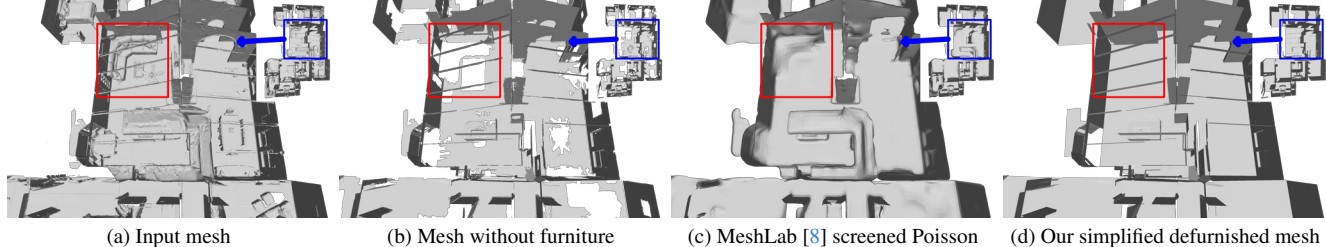

(a) Input mesh      (b) Mesh without furniture      (c) MeshLab [8] screened Poisson      (d) Our simplified defurnished mesh

Figure 3. **Hole filling comparison** between MeshLab's screened Poisson hole filling [8] and our proposed simplified defurnished mesh method. Poisson re-meshing tends to warp surfaces between floors and walls, while they do not interfere in our SDM.

ing any gaps using plane extension. This technique exploits the planar approximation of the mesh to seamlessly extend neighbouring planes and close the gaps left by the removed furniture faces. Additional heuristics-based methods are leveraged after this process to ensure preservation of key features of this mesh, such as doorways. It is worth noting that the implementation of this step may be highly application-dependent, since different features of the mesh may need preservation, removal, or simplification. Figure 3 shows a comparison with standard screen Poisson hole filling in MeshLab [8], which leads to warped surfaces in place of the removed furniture faces, while our SDM keeps planes such as walls and floors flat. Please refer to the supplementary material for a zoomed-in version.

### 3.3. Control Image Generation

Using the camera poses associated with the panorama images and the SDM, we generate depth and normal images and extract Canny edge maps [4] from these images. These Canny edge maps (see example in Figure 2) serve as control inputs for a CN inpainting model. The depth and normal edge images provide geometric guidance, ensuring that the inpainting process maintains the scene's structural integrity. They also contain information that may not be available from a particular view, due to obstruction by the furniture to be removed.

### 3.4. Panorama Image Inpainting

The original panoramas are inpainted using a CN model, guided by the generated depth and normal edge control images. This process effectively removes the furniture from the panorama images while preserving the surrounding scene context. We opt for the Canny edge CN flavour applied to normal/depth images, as it captures fine geometric structures without relying on predefined semantics, making it more adaptable across diverse image domains.

**Vanilla Canny ControlNet** We first evaluate off-the-shelf Canny edge CN weights, *thibaud/controlnet-sd21-canny-diffusers*. We find these weights do not perform well on panorama images, as indicated in Figure 4.

**ControlNet Fine-Tuning** In order to improve quality, we fine-tune the Canny edge CN inpainter on a dataset of 50,000 unfurnished panoramas and corresponding Canny edge maps generated using the approach in Section 3.3. We chose unfurnished panoramas based on performing furniture segmentation across our data and selecting images with no pixels belonging to any of the furniture classes. Given that we want to inpaint "empty room" content, the unfurnished images are already appropriate ground truth targets. To simulate the removal of irregular objects, we employ a composite mask generation technique. This method iteratively constructs a binary mask by superimposing multiple circular regions. For each mask, the number of circles, their radii, and their centre locations are randomly sampled within predefined ranges. This process results in a mask with a complex, irregular shape, mimicking the removal of arbitrary objects from an image. The generated mask, along with the masked input image and Canny control image, are then used as input to the inpainting fine-tuning process. We train this model to convergence based on a $80\% - 10\% - 10\%$ split, picking the checkpoint which maximises PSNR on the validation set.

**ControlNet Inference** During inference, we use the CN inpainter in conjunction with off-the-shelf SD weights. We evaluated SD 2.0 weights, but due to issues with hallucinations, we opted instead for a set of weights fine-tuned following the approach of Slavcheva *et al.* [43] on a dataset of perspective images of unfurnished rooms and their corresponding, virtually-staged counterparts.

### 3.5. Super-Resolution and Blending

We apply the super-resolution network RealESRGAN [55] to upsample the inpainted panorama images to their original resolution (a factor of four). Using the pre-trained weights, the result is generally of an acceptable quality. However, in areas with natural texture (such as wood grain and stone), patterned fabrics, or very high detail (such as carpets), the result is overly smooth and appears artificial. To restore the missing detail, we introduce an image contrast loss using a Laplacian of Gaussian (LoG) operator. We apply the LoG

to the predicted and target images individually and then take the absolute difference of these images as the final loss. Overall, this results in sharper detail, improved natural textures and more realistic looking imagery. The LoG loss also introduces some high-frequency artefacts in low-frequency areas, for example on solid colour walls.

We eliminate the majority of these artefacts by introducing a novel loss we name *FFTMax*. Given the predicted and target images $I_P$ and $I_T$, let $X_P = \text{FFT}(I_P)$ and $X_T = \text{FFT}(I_T)$, then:

$$L_{\text{FFTMax}}(x) = \begin{cases} \left( \frac{(X_P(x) - X_T(x))}{X_T(x)} \right)^2 & \text{if } X_P(x) > X_T(x) \\ 0 & \text{otherwise,} \end{cases}$$

where $x$ is an image coordinate. This loss penalises only where the predicted value is greater than the target value and suppresses the addition of high-frequency content.

Finally, blending is performed following [43] to seamlessly integrate the inpainted regions with the rest of the image, minimising any visual artefacts.

### 3.6. Mesh Texturing

The final defurnished panorama images are used to retexture the SDM. This process effectively transfers the defurnished appearance from the panorama images onto the 3D mesh. Importantly, this allows holes in the textures created during the mesh defurnishing process to be filled.

### 3.7. Output and Resources

The pipeline's output consists of a set of defurnished 360° panorama images and a corresponding defurnished textured mesh. This output represents a complete defurnished scene, ready for further applications or visualisations.

**Datasets** For comparisons with existing work, we utilise publicly available datasets such as Matterport3D [6] and ScanNet [10]. To enhance our model's performance, we generated a large-scale dataset of unfurnished indoor environments, including 50,000 equirectangular panoramas and corresponding Canny edge maps, specifically designed for fine-tuning CN. For the base SD model, we assembled a collection of 20,000 unfurnished perspective images of indoor spaces. These were then augmented with realistic synthetically generated furniture, incorporating accurate illumination and shadows. This process, while broadly inspired by existing defurnishing methodologies [43], emphasises photorealism through detailed lighting and shadow integration, similar to techniques used in recent object manipulation studies, such as those that add/remove physical objects at capture time [61].

**Inference Runtime** For a scene containing 30 panoramas and a corresponding textured mesh (*e.g.* the space from Figure 1), our pipeline takes around 10 minutes, split roughly evenly between image and mesh processing. Specifically, on a *g5.xlarge* instance (4×vCPU, 16GB RAM, A10G GPU) it takes approximately 3s per image for semantic segmentation and 7s per image for CN inference, super-resolution, and blending, totalling approximately 5 minutes. The remaining 5 minutes is taken up by the mesh simplification and defurnishing, canny edge generation, and texturing of the SDM. A scene containing 120 panoramas takes approximately 40 minutes, of which 4 are spent on segmentation, 12 on image defurnishing, and 24 on remaining steps.

## 4. Results

In this section, we analyse the properties of our method via ablation studies and compare to related techniques.

### 4.1. Ablations

**CN + Structural Prior vs Base SD** To evaluate the impact of the SDM geometric prior and CN-based inpainting on the final defurnishing result, we conducted an ablation study using a dataset of 700 equirectangular panoramas from various unfurnished residential spaces. For each image, we generated random masks, simulating furniture removal, and corresponding Canny control images derived from our SDM, as described in Section 3. We then compared three inpainting approaches: a) base SD inpainting, b) CN inpainting with off-the-shelf Canny weights (CN Canny thibaud), and c) CN inpainting with our fine-tuned weights (CN Canny ours). We assessed the quality of the defurnished results against the original images using objective metrics (MSE, PSNR) and perceptual metrics (SSIM [56], LPIPS [69], JOD [26]), both globally and within the masked

Table 1. **Quantitative comparison** between the ground truth unfurnished images and inpainting results obtained using base SD inpainting and CN inpainting with SDM control.

| Metric | SD | CN Canny thibaud | CN Canny ours |
|---|---|---|---|
| MSE (↓) | 0.009 | 0.008 | **0.007** |
| PSNR (↑) | 21.163 | 21.922 | **22.618** |
| SSIM (↑) | 0.848 | 0.852 | **0.854** |
| LPIPS (↓) | 0.118 | 0.106 | **0.092** |
| JOD (↑) | 6.080 | 6.297 | **6.512** |
| MSE (Masked) (↓) | 0.009 | **0.007** | **0.007** |
| PSNR (Masked) (↑) | 21.231 | 22.090 | **22.751** |
| SSIM (Masked) (↑) | 0.906 | **0.912** | 0.910 |
| LPIPS (Masked) (↓) | 0.098 | 0.085 | **0.077** |
| JOD (Masked) (↑) | 6.243 | 6.491 | **6.611** |

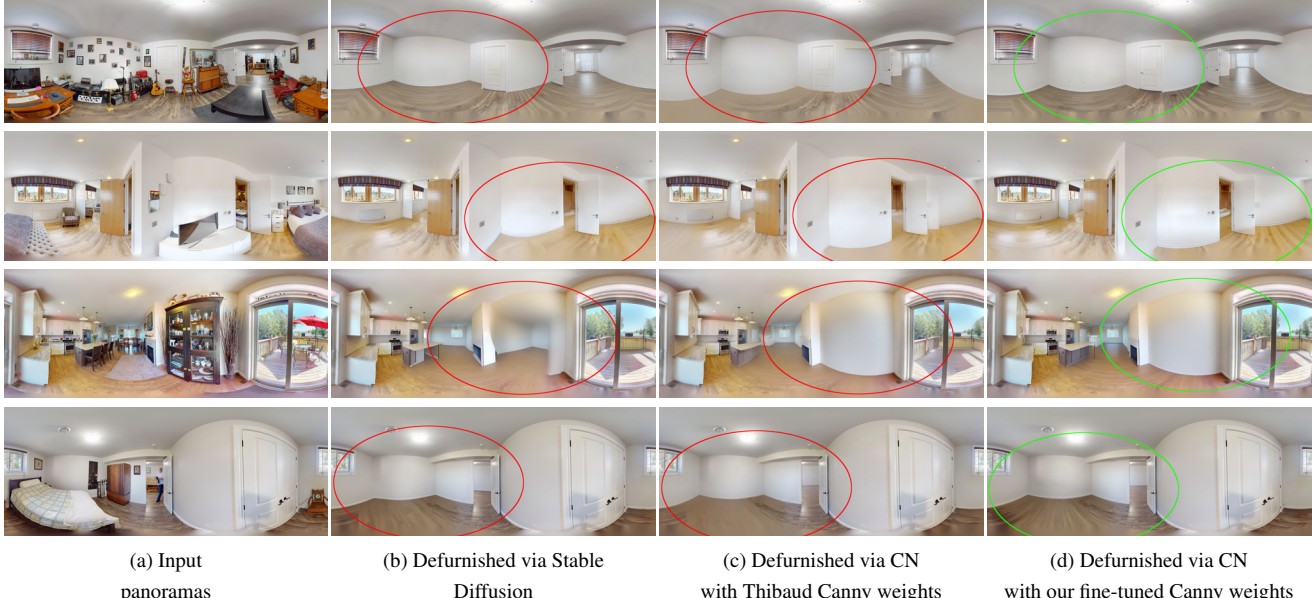

|           |           |           |           |
|-----------|-----------|-----------|-----------|
| (a) Input panoramas | (b) Defurnished via Stable Diffusion | (c) Defurnished via CN with Thibaud Canny weights | (d) Defurnished via CN with our fine-tuned Canny weights |

Figure 4. **Ablation of control method used to guide defurnishing.** Plain SD inpainting often results in warped, unrealistic geometry, such as the wall-floor fusion on the first two rows. The use of Canny edge guided CN makes the inpainting process follow the underlying structure, but off-the-shelf weights tend to hallucinate new rooms when removing large wardrobes (see the third row), while our fine-tuned weights preserve the the wall structure correctly.

regions (denoted as "Masked"). The quantitative results are presented in Table 1, and a qualitative comparison is shown in Figure 4. Please note that while post-processing is used to improve the final result, all metrics are calculated before super-resolution or blending are applied.

Our results demonstrate that incorporating a geometric prior through CN significantly improves inpainting compared to vanilla SD, as evidenced by all evaluation metrics. Fine-tuning CN on panoramic images, random masks, and Canny edge maps derived from the SDM further enhances performance. Interestingly, the off-the-shelf Canny CN exhibited a slightly higher SSIM within the masked regions compared to our fine-tuned version. This marginal difference may stem from the off-the-shelf model's training on precise image-based Canny edge maps [68], while our fine-tuned model uses SDM-derived edges, which might introduce slight inaccuracies. However, perceptual metrics like LPIPS and JOD, which better align with human perception, still favour our fine-tuned CN, indicating that it produces more perceptually accurate and pleasing results overall.

### 4.2. Comparisons

To the best of our knowledge, there are no other methods that deal with the exact problem of furniture removal from 3D scenes, so we compare to other object removal pipelines.

**Radiance Field-based** These methods modify scenes in a two-step process, starting with the creation of an ini-

tial NeRF or 3D Gaussian splatting representation from the input images with furniture, followed by an optimisation process that modifies the initial model. The modification is achieved either via a variant of Score Distillation Sampling [37] that uses a global prompt to gradually update the radiance field, or via iterative dataset updates that use mask-based inpainting, interleaved with radiance field updates.

We found global prompt-based object removal to be unsuccessful for furniture removal in scenes from Matterport3D [6]. With the prompt *remove all furniture from this space*, Instruct-NeRF2NeRF [16] tended to gradually amplify artefacts in the initial NeRF, without modifying furniture. Instruct-GS2GS [51] was more successful at object removal, however, it was not spatially precise - regardless of which objects the prompt specified, it always removed certain objects and kept others. Please refer to the supplementary material for visualisations.

Techniques that rely on inpainting-based dataset updates were more suitable for furniture removal. In Figure 5 we compare to Nerfiller [58], which we modified to start from a depth-guided NeRF model, as we found this to produce better geometry and fewer artefacts than RGB-only NeRF. Note that we train and render on perspective images and only convert to panoramas for visual comparison here. We tested different mask dilation sizes and chose the best result for each scene. Additionally, we ran on entire multi-room spaces and with a NeRF for each room - the results were not markedly different, here we show whole-space results,

|  (a) Original | (b) Defurnished with Nerfiller [58] | (c) Defurnished with our method |

Figure 5. **Defurnishing comparison with a radiance field based method** Nerfiller [58] on a small (180 input images, top) and a big (366 images, bottom) space from Matterport3D [6]. Light reflections and shadows on walls mislead Nerfiller to generate objects that can cause these effects, while our method is trained to be robust to them. When Nerfiller inpaints objects successfully, it tends to leave remnant volumetric density, which appears as blur in images and blobs in meshes, while we output high-resolution images and clean meshes.

while per-room results are in the supplementary material.

The resulting panoramas show that, especially for large objects, Nerfiller is tricked by shadow remains not covered by the inpainting masks and hallucinates objects similar to the inputs. For smaller objects that are well-covered by the masks, the generated appearance is right, but as this is a volumetric approach, it does not manage to fully remove the density that was concentrated to represent the object, which results in nearly transparent points that look like blur, ultimately creating a lower-resolution output than our method.

Lastly, radiance fields are not designed to represent geometry accurately. We can extract meshes from them via Poisson surface reconstruction on thresholded density, however, this yields many spurious points where there should only be empty space, and thus blobby meshes. To obtain a quantitative indication of their precision, we design a synthetic experiment whereby objects from Objaverse [14] are inserted into 3D models with no furniture. The root mean squared error of our SDM compared to the ground-truth unfurnished model is 2.3 cm, while that of Nerfiller is 24.1 cm, an order of magnitude larger. Image metrics and more details can be found in the supplementary material. This experiment confirms that our method is more suitable for downstream tasks that require accurate output geometry.

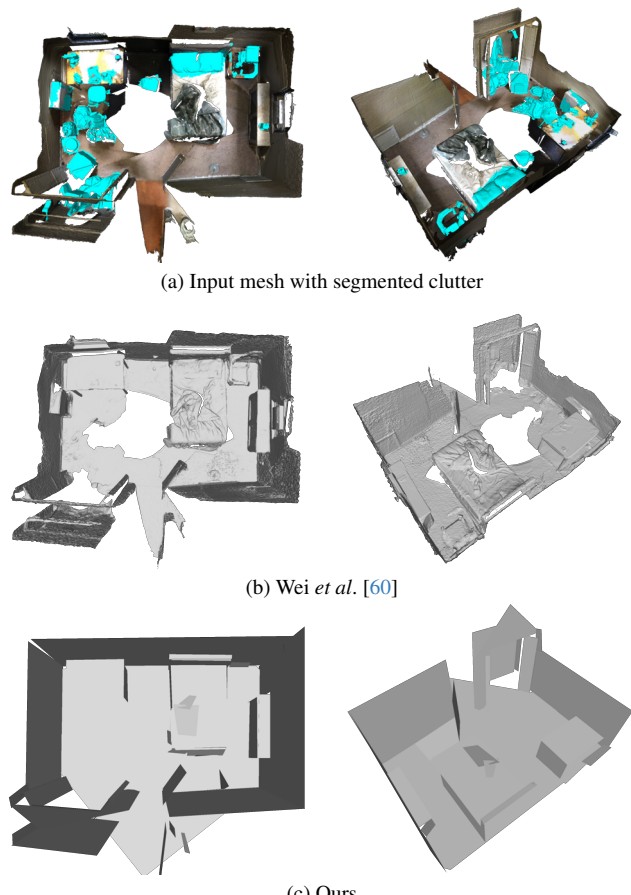

(a) Input mesh with segmented clutter

(b) Wei *et al.* [60]

(c) Ours

Figure 6. **Declutter comparison.** We modify our method for decluttering instead of full defurnish. We achieve cleaner, smoother surfaces than the RGB-D inpainting method of Wei *et al.* [60].

**Mesh clutter removal**   To the best of our knowledge, the only other method that deals with a similar task is the clutter removal work of Wei *et al.* [60]. Their code is not publicly available, so we compare to their final mesh result on ScanNet [10] scene 699, which the authors kindly provided. Please note that we modify our method for this experiment, thus some of our design choices are violated, *e.g.* relating to the water-tightness of the output mesh, as ScanNet scenes do not have ceilings. We also tested Nerfiller on this data, but due to the poorer quality depth and poses, the mesh is too blobby and we only include it as supplementary material. The results are shown in Figure 6. We highlight that our method succeeds at the clutter removal task, which it was not designed for. Our mesh is more complete - it even closed the hole in the input mesh's floor. The mesh of Wei *et al.* tends to have very uneven surfaces where clutter was removed, *e.g.* on the desk, floor, and cupboard, while our mesh is cleaner. Therefore, our result is more suitable for use in further applications, such as architectural processing.

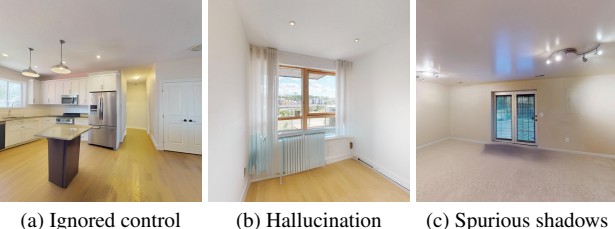

(a) Ignored control    (b) Hallucination    (c) Spurious shadows

Figure 7. **Failure case** examples. *(a) Ignored control signal:* The kitchen island is largely removed despite the existence of corresponding Canny edges. *(b) Hallucination* of a radiator after furniture removal. *(c) Spurious shadows:* The shadow of a sofa is not fully removed. Input images can be found in the suppl. materials.

### 4.3. Limitations and Future Work

While our method makes a leap forward in 3D scene defurnishing by incorporating geometric consistency via CN, it may still suffer from object hallucinations that are inherent to SD [43]. As Figure 7 shows, sometimes issues in the SDM, such as faces left over during furniture removal, may also cause hallucinations. Furthermore, when the input mesh is too complex and hard to simplify, the extracted Canny edges may not be entirely clean, *i.e.* the control signal becomes misleading. Finally, when we are removing furniture that occludes a certain region in one view, if this region is present in another view, the control signal may be insufficient to ensure the geometry is preserved in the inpainted version of the occluded view, leading to view inconsistencies. While radiance fields based methods are inherently view-consistent, we have demonstrated that they are incapable of matching the same image quality as our method. Further research is needed to achieve high-resolution view-consistent results. One intriguing direction was set by MVDiffusion [47], where the view consistency is achieved through attention layers in a transformer architecture. However, this kind of approaches need to process multiple images simultaneously, *i.e.* they can currently only operate at lower resolutions.

### 5. Conclusion

This paper introduced a novel method for jointly defurnishing 3D scenes and their corresponding panoramic images. Our approach leverages the geometric information from a simplified mesh to guide the inpainting process, ensuring consistent results. We demonstrated the effectiveness of our approach through extensive experiments, highlighting its ability to handle complex, real-world environments.

This work contributes a robust solution for defurnishing, with implications for virtual staging and 3D scene understanding. Future work will explore incorporating more sophisticated inpainting techniques and extending the approach to other 3D scene manipulation tasks.

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
