# Defurnishing with X-Ray Vision:
# Joint Removal of Furniture from Panoramas and Mesh

## Supplementary Material

## 6. Results

We include higher-resolution versions and more examples for several of the figures in the main paper.

Figure 8 shows perspective images corresponding to our inpainted panoramas for easier evaluation of qualities like line straightness.

Figure 9 adds more viewpoints of our comparison to screened Poission hole filling in MeshLab from Figure 3.

Figure 10 shows larger versions of our failure case examples from Figure 7, together with the respective input images.

## 7. Radiance Fields Methods

Here we add details and results from our experiments with methods that rely on radiance fields for object removal.

We ran these experiments on Matterport3D [6] and ScanNet [10] data. For Matterport3D we show a small studio apartment, consisting of 180 images, and a larger multi-room house, consisting of 366 images.

### 7.1. Nerfiller [58]

We use the authors' *nerfstudio* [46]-based implementation, which runs 30 thousand steps to create an initial NeRF and 30 thousand steps to inpaint masked regions. The default method for training the initial NeRF is *nerfacto-nerfiller*, which is very similar to standard *nerfacto* and only uses poses RGB images as input. We found that in these indoor spaces *depth-nerfacto*, which uses posed RGB and depth images, creates a better initial NeRF with less floater artifacts, as shown in Figure 11. Therefore, we use the depth-based NeRF variant as initialization in our experiments.

Meshes and some panoramas on Matterport3D data are shown in Figure 5 of the main paper, while and Figure 12 shows more panoramas for each space. Note that we train, inpaint and render Nerfiller and any other radiance fields on $512 \times 512$ perspective images, as they are intended to be used, and only convert the outputs to panoramas afterwards for easier comparison with our results.

The figures show both the smaller (top) and larger (bottom) spaces. We trained a single NeRF/Nerfiller for the small space, but for the large space we tried both training on the entire space and separately on the living room and bedroom. The main paper shows results from training one model per space. Here the bottom of Figure 12 shows panoramas that were obtained by training one model for the living room (first two images below the mesh snapshots) and one model for the bedroom (next two rows). The living room dataset contains 114 images, while the bedroom contains 30 (the inpainting step of Nerfiller requires the image number to be a multiple of 4, so we dropped two floor-facing frames for a total of 28 during inpainting). As mentioned in the main paper, the results are not markedly different, and arguably slightly worse with a separate model per room. Therefore, poor results cannot be attributed to insufficient NeRF capacity and are inherently related to the sensitivity to shadows and light reflections of off-the-shelf Stable Diffusion inpainting.

Additionally, we show the mesh extracted from Nerfiller's inpainted result via Poisson surface reconstruction on the ScanNet test scene in Figure 13. Due to the poor depth and inaccurate poses in this dataset, and inpainting process that creates blobs around volumetric density, the mesh is unrecognizable. The figure also shows a few frames rendered from the inpainted model, demonstrating a reasonable, but not seamless, inpainting result.

### 7.2. Instruct-NeRF2NeRF [16], Instruct-GS2GS [51]

Similarly to Nerfiller, Instruct-NeRF2NeRF requires an initial NeRF, which is then trained for 15 thousand steps with a prompt. Therefore, we again start from *depth-nerfacto* for higher accuracy and fewer floaters in the representation that will be modified. As shown in Figure 11, 3D Gaussian splatting has fewer artifacts than both NeRF variants on this data, so we also test Instruct-GS2GS.

We tested Instruct-NeRF2NeRF with prompts *remove all furniture from this space*, *empty room*, *Show this as an empty room without furniture. Keep the current floor, walls, ceiling, windows and doors.*, *i.e.* we tried prompts for furniture removal of different length and specificity. For each prompt we followed the recommended practice of verifying that inpainting on a few images from our dataset results in reasonable results via the Instruct-Pix2Pix HuggingFace page (https://huggingface.co/spaces/timbrooks/instruct-pix2pix). We observed the same trend for all of them, which is demonstrated in Figure 14: the representation progressively gets more filled with floaters and discolored over iterations. Furniture is not removed, as we can still see outlines of the couch, TV, bed, cupboards. Structure is not kept, as we clearly see that the floor and kitchen built-ins get equally discoloured. It seems that the global prompt is only leading to amplification of the floaters present in the initial NeRF.

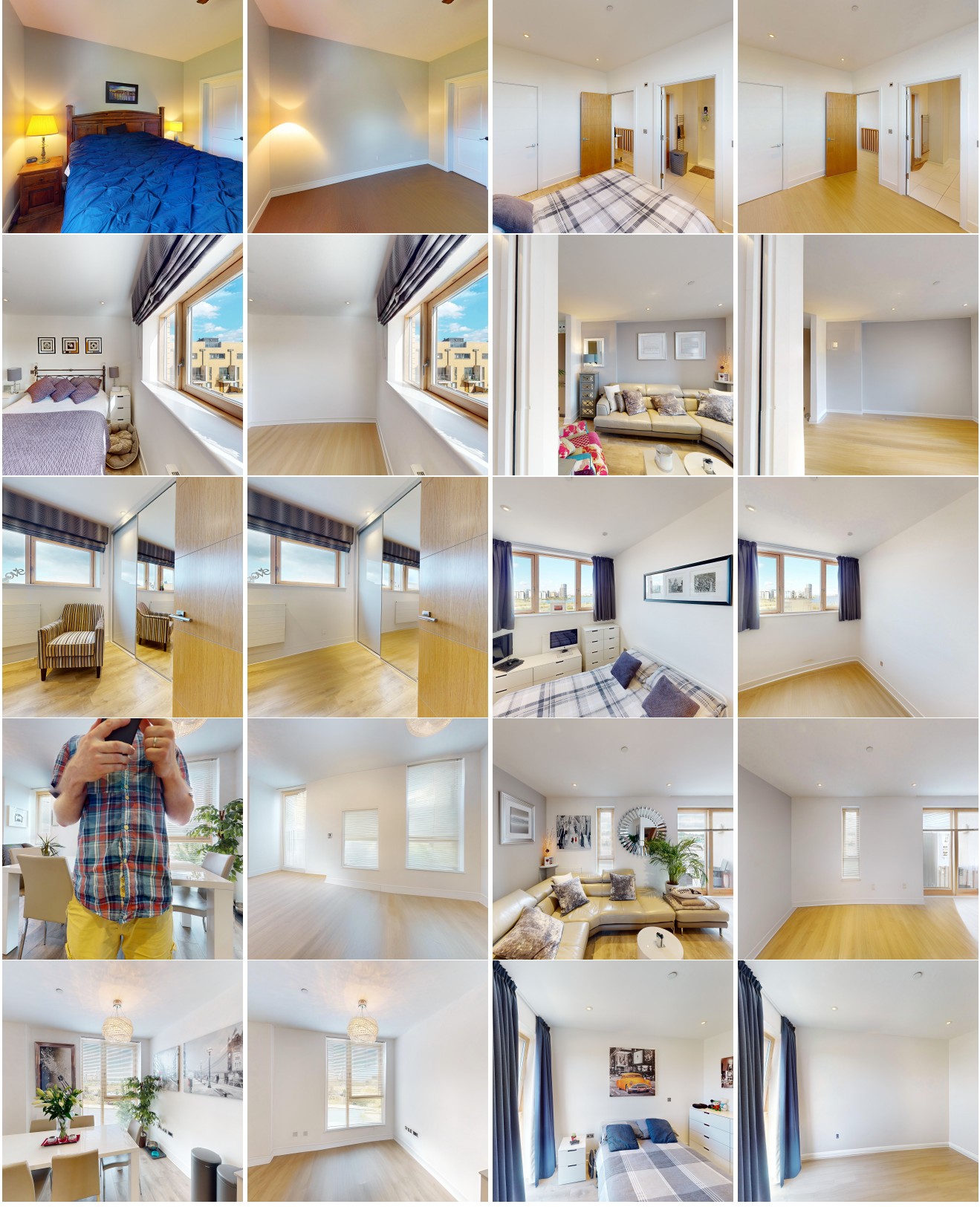

Figure 8. **Pairwise comparisons of perspective renders** of furnished inputs and results defurnished using our pipeline. This projection highlights some remaining issues with straight wall/floor/ceiling edges, which do not always get resolved, even when using Canny ControlNet.

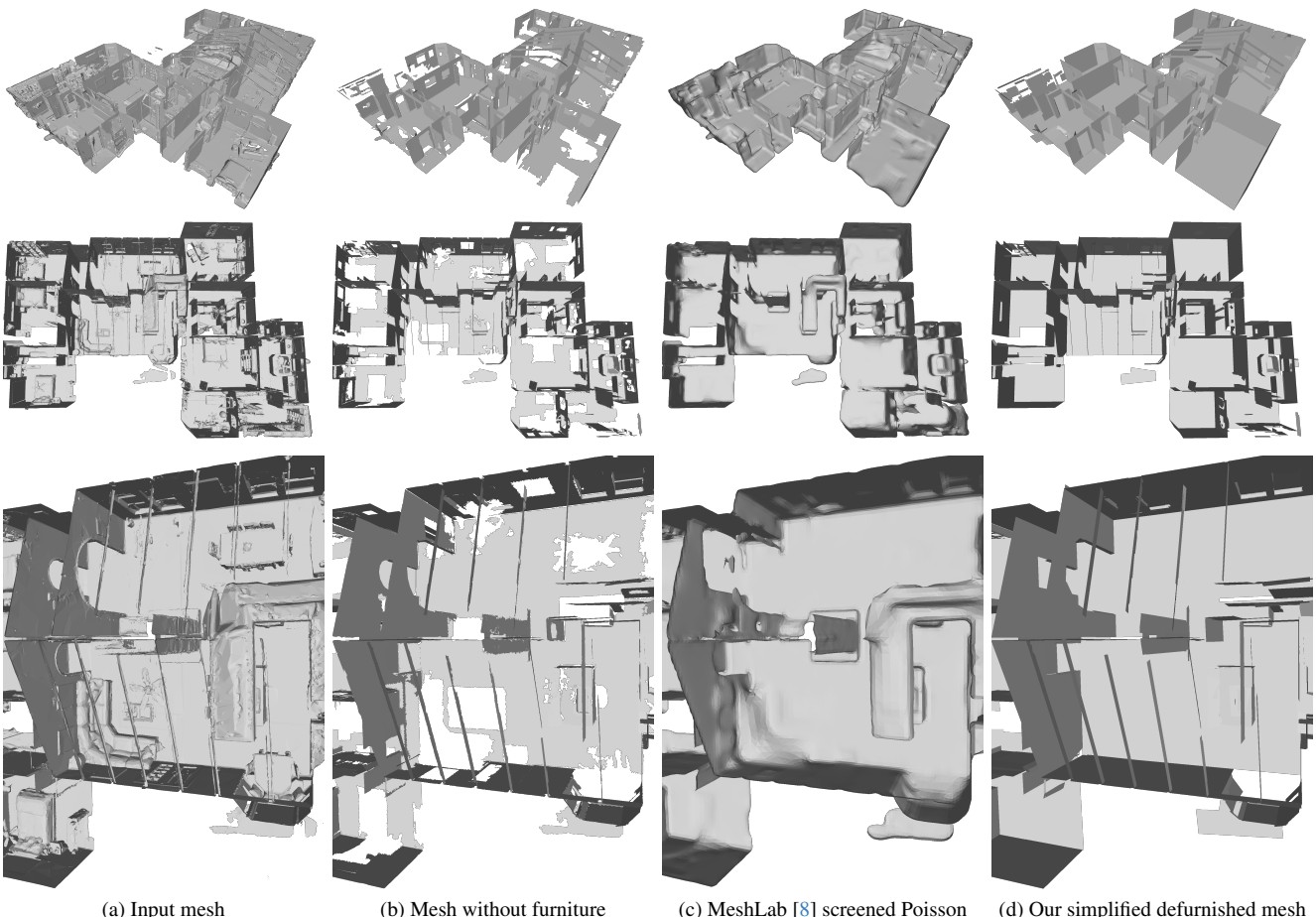

(a) Input mesh     (b) Mesh without furniture     (c) MeshLab [8] screened Poisson     (d) Our simplified defurnished mesh

Figure 9. **Hole filling comparison** between MeshLab's screened Poisson hole filling [8] and our proposed simplified defurnished mesh method. Poisson re-meshing tends to warp surfaces between floors and walls, while they do not interfere in our SDM.

Therefore, we try the variant of the method based on the cleaner Gaussian splatting representation, as shown in Figure 15. We find this method to be better at scene modification and in particular object removal, however, it is not spatially precise. For instance, the word *remove* causes removal of items such as the table, sofa, bed, coffee machine, regardless of whether the prompt asks to remove all furniture, just the sofa, or just the TV. Notably, with the prompt *remove the TV*, the TV remains in the scene, while all the aforementioned objects get removed. Thus we experimented with more localized scene modification. Similarly, the prompt *make the sofa green* successfully makes the sofa green, but also turns the walls, sink, and kitchen island top, slightly green, *i.e.* this kind of modification is also not precise. We also noticed that turning objects into geometrically similar objects works, *e.g.* a horse statuette into a zebra statuette, but removal, even if successful, typically leaves artefacts as observable in Figure 15. Note that for scene modification the default 7.5 thousand steps recommended by the authors were sufficient, however, for object removal at least

20 thousand steps were necessary to see the majority of the object's geometry removed. All images here are rendered after 30 thousand steps.

With this we conclude that global SDS-based object removal is not sufficiently precise for our purposes.

# 8. Quantitative Evaluation on Synthetic Data

To evaluate the performance of our method against Nerfiller, we conducted experiments using synthetically furnished 360° panoramas and corresponding mesh. We began with a dataset of unfurnished 3D spaces, represented as meshes and corresponding panos. To simulate furnished environments, we procedurally insert 3D furniture objects, and their approximate shadows, into both the mesh and the associated panos. This process creates pairs of "furnished" meshes and panos. Subsequently, we apply the same defurnishing techniques as described before - vanilla Stable Diffusion (SD) inpainting, ControlNet (CN) inpainting with two sets of Canny edge weights (Thibaud's and ours), and

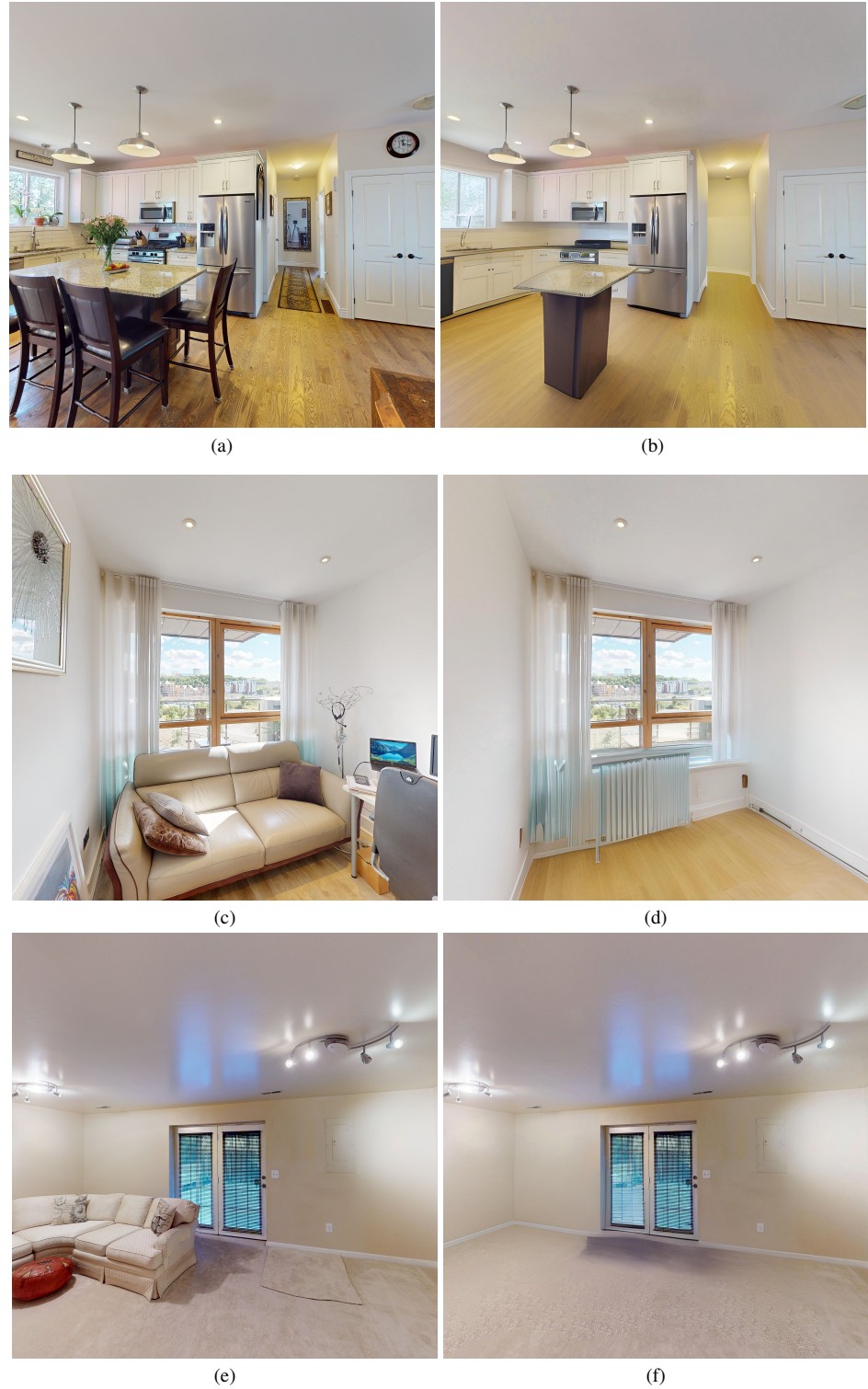

Figure 10. **Failure case** examples. *Ignored control signal:* The kitchen island in a) is largely removed in b), despite the existence of corresponding Canny edges. *Hallucination:* after removing the furniture in c) a radiator is hallucinated in d). *Spurious shadows:* the shadow of the sofa in e) is not fully removed in f)

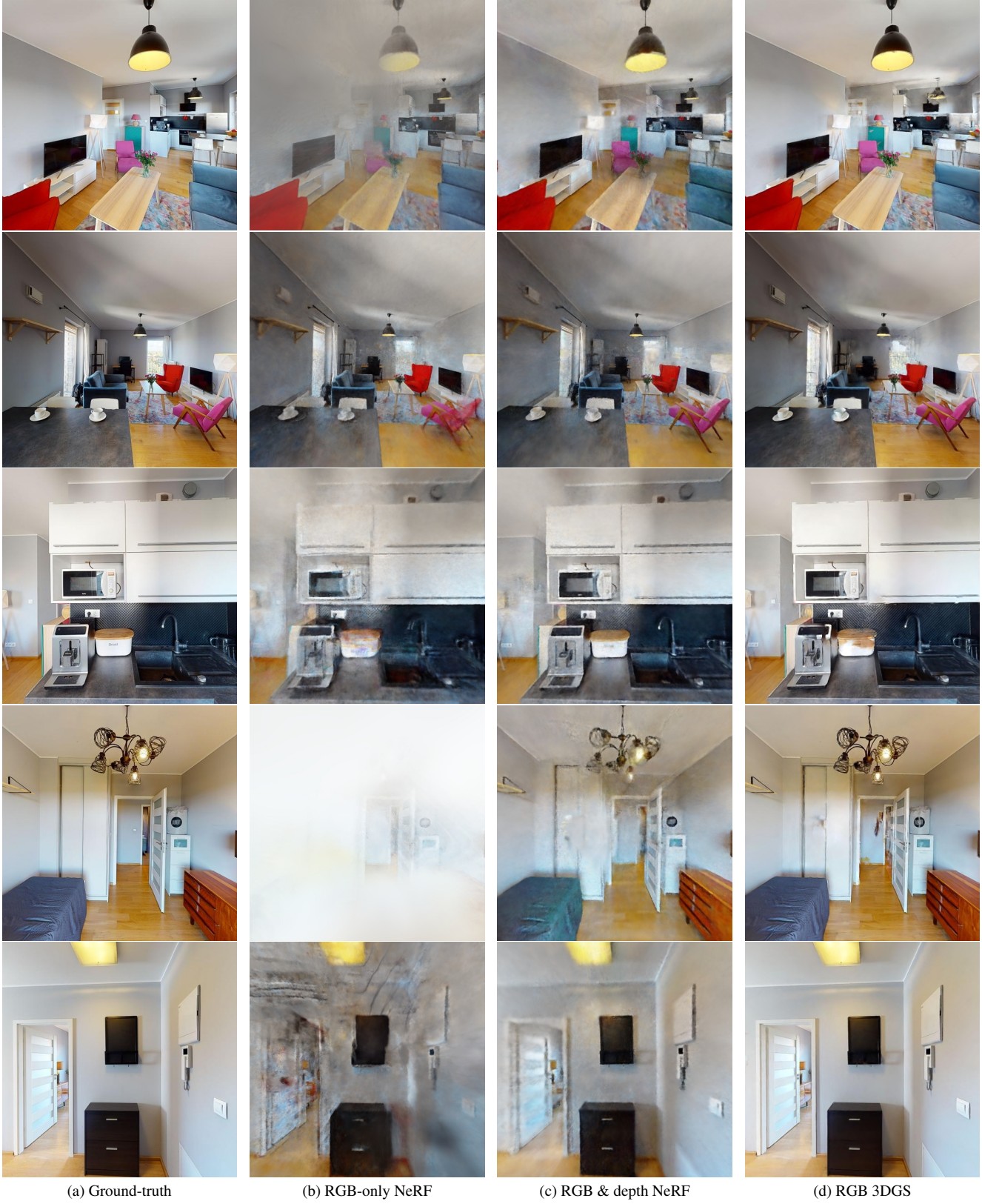

| (a) Ground-truth | (b) RGB-only NeRF | (c) RGB & depth NeRF | (d) RGB 3DGS |

Figure 11. **Radiance field initialization comparison.** Posed RGB-only NeRF (*nerfacto* and *nerfacto-nerfiller*) exhibits more floater artifacts than posed RGB-D NeRF (*depth-nerfacto*), while posed RGB-only 3D Gaussian splatting (*splatfacto*) is cleanest.

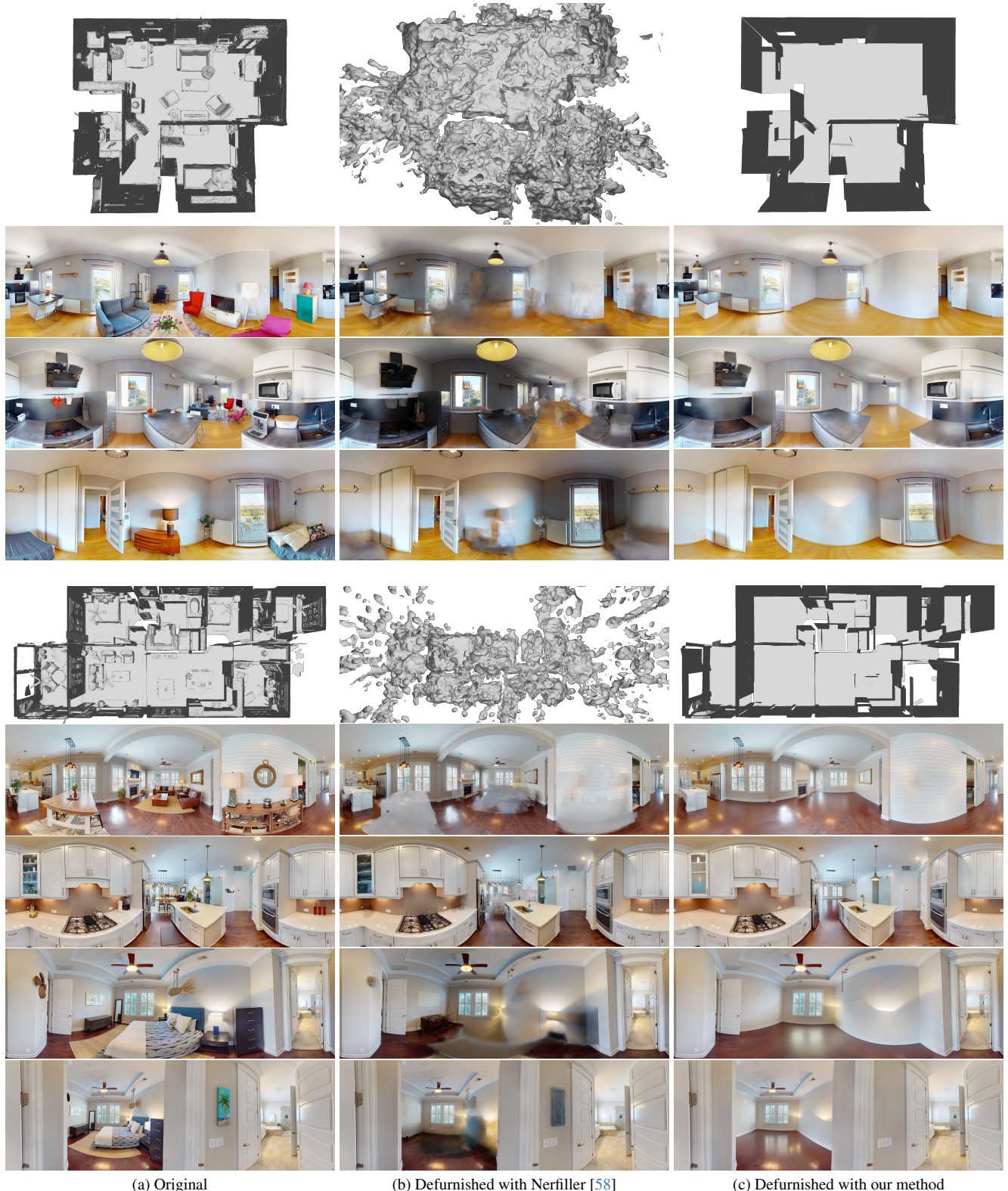

(a) Original        (b) Defurnished with Nerfiller [58]        (c) Defurnished with our method

Figure 12. **Defurnishing comparison with a radiance field based method** Nerfiller [58].

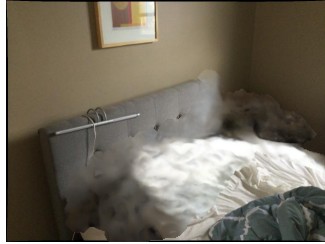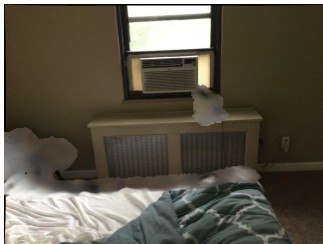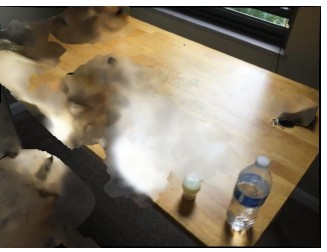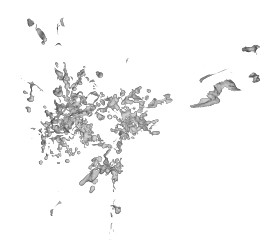

Figure 13. **Inpainted frames and mesh** extracted via Poisson surface reconstruction on Nerfiller's inpainted model on the ScanNet scene from Section 4.2.

Table 2. **Quantitative comparison on synthetic data** between ground truth and inpainting results. Our proposed method (CN Canny Ours) achieves superior inpainting performance compared to vanilla SD and CN Canny Thibaud, as indicated by the metrics, especially within the masked region. Minor differences in overall image metrics may reflect Nerfiller's higher global image quality.

| Metric | SD | CN Canny thibaud | CN Canny ours | Nerfiller |
|---|---|---|---|---|
| MSE (↓) | 0.006 | 0.006 | 0.006 | **0.005** |
| PSNR (↑) | 26.066 | 25.661 | **26.732** | 26.538 |
| SSIM (↑) | 0.787 | 0.783 | 0.790 | **0.803** |
| LPIPS (↓) | 0.058 | 0.063 | **0.053** | 0.095 |
| JOD (↑) | 8.018 | 7.933 | 8.177 | **8.192** |
| MSE (Masked) (↓) | 0.001 | 0.001 | 0.001 | 0.001 |
| PSNR (Masked) (↑) | 34.343 | 34.128 | **35.161** | 33.040 |
| SSIM (Masked) (↑) | 0.988 | 0.988 | 0.988 | 0.988 |
| LPIPS (Masked) (↓) | **0.004** | 0.005 | **0.004** | 0.009 |
| JOD (Masked) (↑) | 8.970 | 8.961 | **9.003** | 8.691 |

Nerfiller, to the furnished panos and mesh. Finally, we quantitatively compare the defurnished results against the original, unfurnished panos using the same metrics as in Table 1. This comparison allows us to assess the effectiveness of each defurnishing method in reconstructing the original unfurnished scene. We also performed a comparison only inside the masked region where the furniture was added, to evaluate the performance of each method on the inpainted area specifically. The results of this experiment can be found in Table 2.

The quantitative comparison on synthetic data reveals several key insights into the performance of different defurnishing methods. Overall, our proposed method consistently demonstrates strong performance, particularly within the masked inpainting region, where it outperforms all other approaches, indicating superior reconstruction accuracy. Thus, our method excels at the key objective - recreating the area where the furniture was removed.

When considering the entire image, CN Canny Ours still performs well, achieving the highest PSNR and a competitive LPIPS. Nerfiller demonstrates the highest SSIM and JOD across the entire image. This suggests that while CN Canny Ours excels in inpainting accuracy, Nerfiller may produce a more globally consistent and visually appealing result. This could be due to the nature of the Nerfiller method, which is designed to produce a full 3D reconstruction and then render a 2D image. The vanilla SD method performs the worst in most global image metrics.

Comparing the two ControlNet methods, CN Canny Ours consistently outperforms CN Canny Thibaud, indicating that our optimized weights contribute to improved defurnishing performance. In summary, CN Canny Ours provides a strong balance between inpainting accuracy and overall image quality, making it a highly effective defurnishing method. Nerfiller, while potentially less accurate in the inpainting region, produces a high-quality overall image.

The root-mean-squared model error reported in Section 4.2 is calculated as a cloud-to-mesh error, where the cloud is the model we are evaluating and the mesh is the ground-truth unfurnished mesh. The fact that, on avergae, Nerfiller's 3D model is an order of magnirure less accurate than ours is a strong signal that radiance field-based method are currently not suitable for applications that require high metric accuracy of the underlying 3D models.

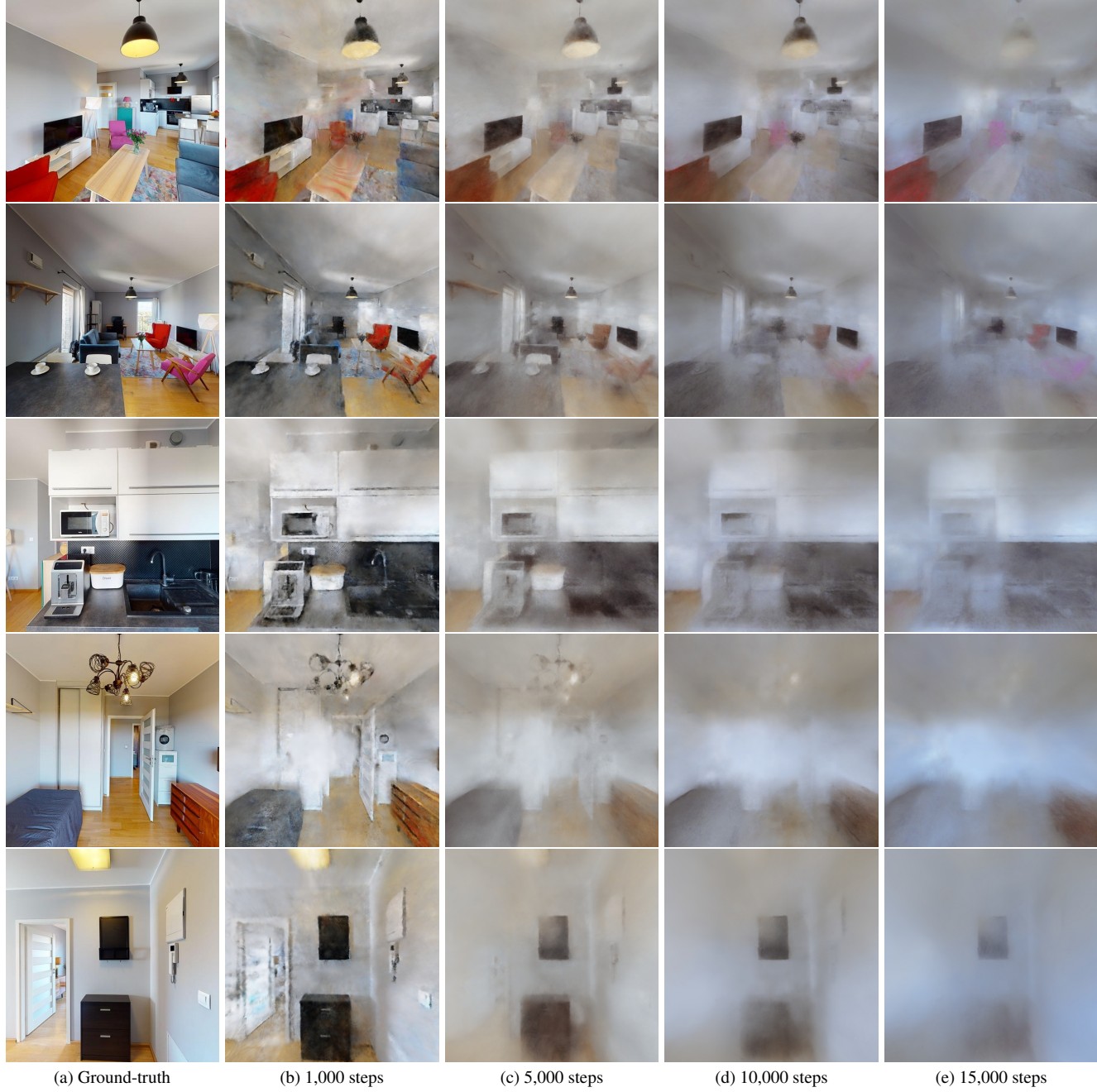

|  |  |  |  |  |
|---|---|---|---|---|
| (a) Ground-truth | (b) 1,000 steps | (c) 5,000 steps | (d) 10,000 steps | (e) 15,000 steps |

Figure 14. **Instruct-NeRF2NeRF** [16] experiments on furniture removal. The prompt used was *remove all furniture from this space*. The modified scene gets progressively blurrier over time, due to amplification of floaters in the initial NeRF.

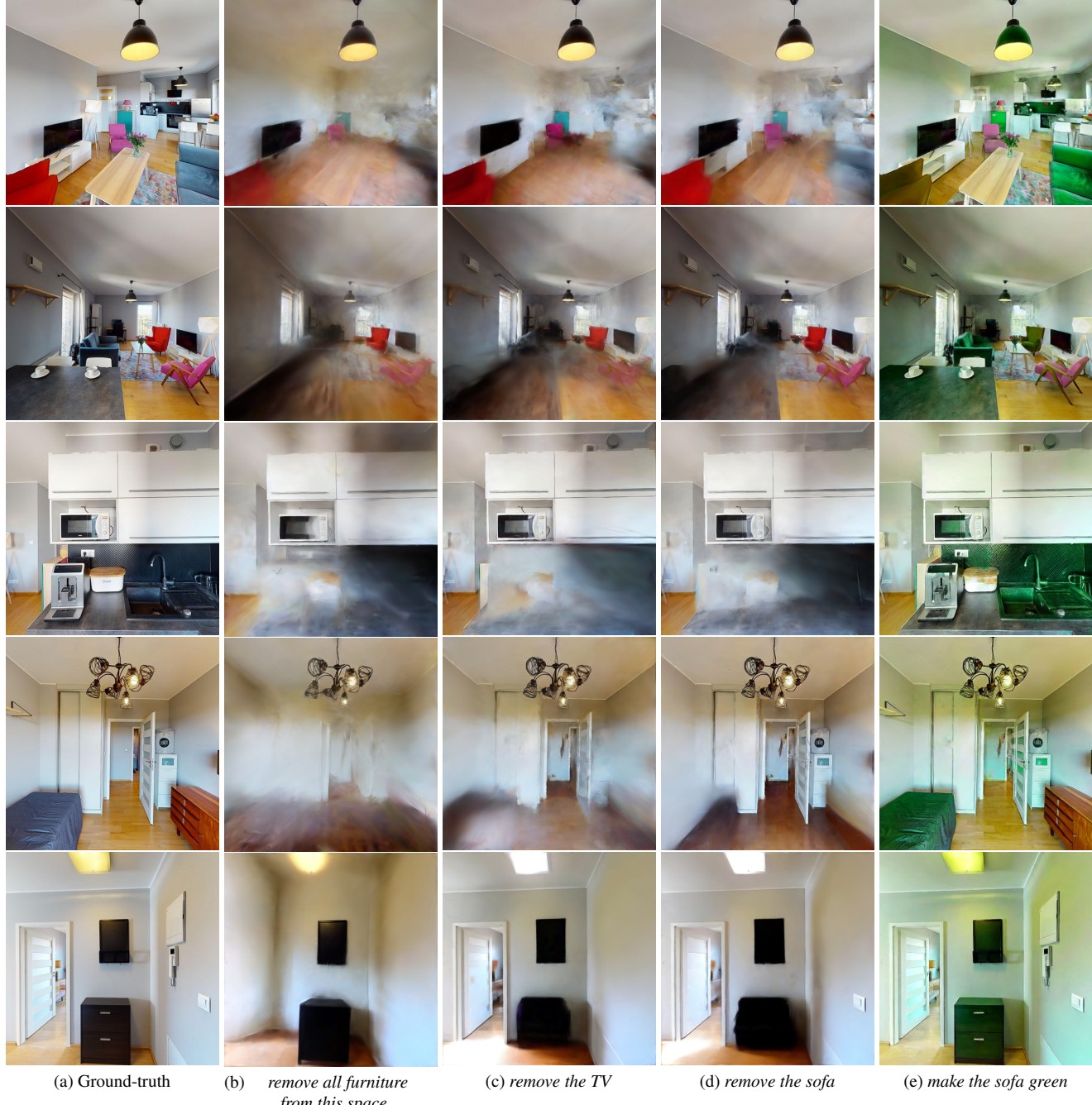

(a) Ground-truth    (b) *remove all furniture from this space*    (c) *remove the TV*    (d) *remove the sofa*    (e) *make the sofa green*

Figure 15. **Instruct-GS2GS** [51] experiments on furniture removal and modification. The prompt used for each experiment is shown in the captions above. While better than Instruct-NeRF2NeRF, Instruct-GS2GS is not sufficiently spatially accurate for our purposes, as evident from the removal of the same objects regardless of the exact prompt in (b), (c), (d), and from the green tinting of other surfaces besides the sofa in (e).