# OpenReview forum: "Defurnishing with X-Ray Vision: Joint Removal of Furniture from Panoramas and Mesh"
_thecvf.com/CVPR/2025/Workshop/CVEU — CVPR 2025_

### Official Review · Reviewer_9ybh · 2025-03-22
**Interesting application of GenAI, decent evaluation effort**

**Rating:** 4
**Confidence:** 3

**Review:**

Authors propose a method that takes in 360 degree panorama images and the corresponding textured mesh of a room, and outputs a defurnished geometry "SDM", which can be used to render an empty room. They utilize the textured mesh to extract depth priors to ground the inpainting process to have better geometrically correct outputs such as straight lines for walls and floors.

# Pros
- Tackles the interesting and practical problem of defurnishing, which has useful downstream applications for digital twins and real estate.
- I like how most of the large models used have been adapted for the specific use case. The authors provided reasonable explanations for all the fine-tuning.
- Thorough numerical and visual comparisons with chosen baselines in the main paper and supplementary.
- I like the idea of incorporating depth priors in general, especially for tasks like inpainting.

# Cons
- I am concerned about the actual usefulness of the proposed method, which requires panorama images *and* the actual mesh geometry of the scene. How often do these room images come with decent geometry, if it's even provided? This method relies on the quality of the input geometry, which I feel might not be available most of the time. (I might be wrong—if so, consider adding some stats or use-case examples in the introduction to contextualize this.)

# Minor Points
- **Additional discussion on related work**: There is one cited work, *"An Empty Room is All We Want: Automatic Defurnishing of Indoor Panoramas"* by Slavcheva et al., that seems highly relevant and worth more discussion than a brief mention in line 254. I looked up this paper and found a [blog post by the team](https://matterport.com/features/defurnish?srsltid=AfmBOopuqu630AB5IR-iP4bbEblUfoJX7UYxq_Ecd0ipIKTZb7ZHBwrW) that includes compelling application examples. Including similar visuals of your own results can help communicate the value of this work—many readers need easy-to-understand, visual applications to appreciate the impact. Authors could also consider adding such examples to their project website.

# Final Remarks
Overall, I think the authors have done a solid job evaluating their method, and it presents an interesting application of generative AI for a real-world, industry-focused task. I would like to see more clarification on the practical availability of the required inputs (panoramas *and* geometry), but I found the approach well-motivated and promising.

---

### Official Review · Reviewer_CMy5 · 2025-03-23
**lacks innovation for using SD to produce higher-quality images from initially partial renders**

**Rating:** 2
**Confidence:** 3

**Review:**

Strengths:

The overall pipeline is clear and performs well for removal furniture from Panoramas and Mesh.

Weaknesses:

1、The article overall lacks innovation. It involves directly editing the mesh, then rendering images from corresponding viewpoints, and using these rendered images as conditions for Stable Diffusion (SD) to generate new images. Essentially, it resembles an A+B type of work, similar to previous efforts that reconstruct from sparse view images. By leveraging the generalization capabilities of SD to produce higher-quality images from initially partial renders. However, there doesn't seem to be any fundamental difference from this type of work.

2、The writing in the paper is not very formal；Is this paper a CVPR paper or a CVEU paper?

3、How is the overall generalization capability of the model? For example, how does it perform on outdoor scenes? Why is the testing only conducted on 360 panoramas? if it can remove furniture from Panoramas and Mesh well, why can not it apply to other scenes?

---

### Official Review · Reviewer_aCWq · 2025-03-26
**Novel Task and Good Preliminary Results; Evaluation can be improved**

**Rating:** 4
**Confidence:** 3

**Review:**

This work tackles the novel task of removing furniture from a 3D scene (comprised of a 3D mesh + panorama image). The paper is well-written, describes the method and limitations clearly, and also includes a good discussion of other related methods (e.g., NeRF-based methods). The mesh results look good; however, I'm not convinced that their panorama inpainting is better than baseline methods (e.g., vanilla SD or CN). Overall, I think this new task and the authors' approach would be useful to the computer vision community.

## Pros:
- Novel Task
- Clear writing
- Good mesh results (planar assumption creates smooth meshes in the 3D output)
- Good discussion of limitations

## Cons:
- __Evaluation__: The authors claim that their panorama inpainting approach is better than two baselines. However, it is difficult to see differences between the baseline outputs and theirs. Specifically, in Figure 4, I don't see artifacts in many of the red-circled regions; they look almost identical to the the authors' output. It would be helpful if the authors provided a zoomed-in perspective view of these panoramas for easier comparison. If the authors' inpainting method does not significantly outperform baselines, I suggest not claiming the inpainting portion as a contribution. Since the author's inpainting approach is a fine-tuned version of the baselines, I recommend discussing these changes as implementation details, rather than a contribution.
- __Related Work__: the RW section covered relevant works in related domains; however, there is no mention of how this work differs from existing work. For each paragraph in RW, I would recommend adding 1-2 sentences at the end summarizing how this work differs from other works in the paragraph.

While there are areas of improvement, overall I enjoyed reading this paper and believe it would be useful to the CVPR workshop audience. Hence, I recommend weak accept.

---

### Decision · Program_Chairs · 2025-03-25

**Decision:**

Accept

**Comment:**

The paper proposes a generative AI-based approach for defurnishing indoor panoramas and textured meshes, effectively utilizing Stable Diffusion and depth priors for realistic results. Reviewers recognized the clear practical value, thorough evaluation, and thoughtful adaptations for the task. Concerns were raised regarding limited methodological innovation and dependency on high-quality geometry inputs.

Despite limited novelty, the practical applicability and robust evaluations justify acceptance. Thus, the paper is accepted. Authors should clarify input availability and enhance discussions on generalization and related works in the camera-ready version.